# Dependence of Successful Airway Management in Neonatal Simulation Manikins on the Type of Supraglottic Airway Device and Providers’ Backgrounds

**DOI:** 10.3390/children11050530

**Published:** 2024-04-28

**Authors:** Takahiro Sugiura, Rei Urushibata, Satoko Fukaya, Tsutomu Shioda, Tetsuya Fukuoka, Osuke Iwata

**Affiliations:** 1Department of Pediatrics and Neonatology, Toyohashi Municipal Hospital, 50 Aza Hakken Nishi, Aotake-cho, Toyohashi 441-8570, Japan; o.iwata@med.nagoya-cu.ac.jp; 2Department of Pediatrics and Neonatology, Shizuoka Saiseikai General Hospital, Shizuoka 422-8527, Japan; rei-urushi@ncnp.go.jp (R.U.); t166294@siz.saiseikai.or.jp (T.S.); t162795@siz.saiseikai.or.jp (T.F.); 3Department of Pediatrics, Hamamatsu University School of Medicine, Hamamatsu 431-3192, Japan; 4Center for Human Development and Family Science, Department of Pediatrics and Neonatology, Nagoya City University Graduate School of Medical Sciences, 1 Kawasumi, Mizuho, Nagoya 467-8601, Japan; sfukaya@med.nagoya-cu.ac.jp

**Keywords:** airway management, health occupations, laryngeal mask, newborn, resuscitation

## Abstract

Supraglottic airway devices such as laryngeal masks and i-gels are useful for airway management. The i-gel is a relatively new device that replaces the air-inflated cuff of the laryngeal mask with a gel-filled cuff. It remains unclear which device is more effective for neonatal resuscitation. We aimed to evaluate the dependence of successful airway management in neonatal simulators on the device type and providers’ backgrounds. Ninety-one healthcare providers performed four attempts at airway management using a laryngeal mask and i-gel in two types of neonatal manikins. The dependence of successful insertions within 16.7 s (75th percentile of all successful insertions) on the device type and providers’ specialty, years of healthcare service, and completion of the neonatal resuscitation training course was assessed. Successful insertion (*p* = 0.001) and insertion time (*p* = 0.003) were associated with using the i-gel vs. laryngeal mask. The providers’ backgrounds were not associated with the outcome. Using the i-gel was associated with more successful airway management than laryngeal masks using neonatal manikins. Considering the limited effect of the provider’s specialty and experience, using the i-gel as the first-choice device in neonatal resuscitation may be advantageous. Prospective studies are warranted to compare these devices in the clinical setting.

## 1. Introduction

Although most newborn infants successfully transition from intrauterine to extrauterine life without special help, approximately one in ten newborn infants require some form of medical assistance before spontaneous breathing is successfully established, and approximately 1 in 100 newborn infants require extensive cardiopulmonary resuscitation [1]. Positive pressure ventilation plays a pivotal role in neonatal resuscitation, given that respiratory failure typically precedes cardiac failure in newborn infants. Ineffective positive-pressure ventilation over prolonged periods could lead to the need for advanced resuscitation, including intubation, chest compressions, and adrenaline administration. Notably, every 30 s delay in initiating efficient ventilation is associated with a 16% increase in the risk of mortality or morbidity [2]. However, the optimal interface for administering positive pressure ventilation during the initial stages of resuscitation remains unclear.

Bag-and-mask ventilation is the most common form of positive-pressure ventilation [1]. Although a correctly executed bag-and-mask procedure offers ventilation comparable to that achieved with a tracheal tube, this technique requires training and experience. Similarly, tracheal tube intubation inherently involves a balance between the potential benefits and risks. A comprehensive registry study encompassing over 2600 intubation episodes in academic neonatal units highlighted significant challenges: severe oxygen desaturation, defined as a reduction of 20% or more from baseline oxygen saturation levels, in 41% of these episodes [3]. Successful intubation on the first attempt occurred in only half of the patients. A multivariable analysis within the registry study revealed that an increasing number of intubation attempts was correlated with a significantly heightened risk of adverse events (adjusted odds ratio [OR] 1.87, 95% confidence interval [CI] 1.63–2.14). Clinicians must consider alternative methods after an initial unsuccessful intubation attempt to avoid escalating the risk of harm through repeated failed attempts.

Laryngeal masks and other types of supraglottic airway devices are increasingly used for neonatal resuscitation. Global practice guidelines recommend the use of supraglottic airway devices for resuscitation of term and preterm newborns delivered ≥ 34 weeks gestation when tracheal intubation is not successful or feasible [1]. Furthermore, a recent study on late preterm and term infants reported that the time to spontaneous breathing was shorter with the use of supraglottic airway devices than with a face mask [4]. Consequently, the current guidelines suggest that supraglottic airway devices may be used in the place of face masks for newborns receiving positive pressure ventilation after delivery if the resources and training are available [5].

Since the introduction of the first laryngeal mask in 1983 [6], many revisions have been made to supraglottic airway devices to improve the safety and efficacy of airway management [7]. While the classical laryngeal mask must be inflated before use, the i-gel (Intersurgical Ltd., Wokingham, Berkshire, UK) has improved utility by replacing the air-inflatable cuff with a gel-filled ergonomic cuff (Figure 1).

Several studies have compared the effectiveness of laryngeal masks and i-gels in anesthetized children and found similar success rates [8,9,10,11,12,13,14,15]. However, studies have reported conflicting findings regarding insertion time. Several studies have demonstrated that the insertion time was shorter with the i-gel than with a laryngeal mask [8,10,14], whereas other studies reported similar [9,11] or longer [12,13,15] insertion times with the i-gel than with a laryngeal mask. Considering that the backgrounds of the patients and providers differed between these studies, these conflicting findings suggest that the effectiveness of airway management using a laryngeal mask and i-gel varies according to the patient type, procedure (resuscitation or anesthesia), and the provider’s background. Furthermore, these studies focused on airway management in children undergoing general anesthesia; therefore, little is known about airway management in newborn infants who need to be resuscitated shortly after birth. With further information on the dependence of successful airway management on the type of supraglottic airway device, background of newborn infants, and experience of resuscitation providers, the quality and efficacy of neonatal resuscitation may be significantly improved.

This study aimed to assess the dependence of successful laryngeal mask and i-gel insertion on the specialty, career, and experience of the resuscitation provider using neonatal simulation manikins.

## 2. Materials and Methods

This was a crossover randomized trial using two types of supraglottic airway devices in two types of neonatal simulation manikin. The study protocol was approved by the Institutional Review Board of Shizuoka Saiseikai General Hospital (#20231601). Participants were recruited from healthcare professionals who are currently working, training, or have previous work experience in the NICU or pediatric wards at Shizuoka Saiseikai General Hospital, which admits approximately 300 newborns to the NICU annually, with around 100 newborns receiving mechanical ventilation. Written informed consent was obtained from all participants.

### 2.1. Devices Used

Two types of supraglottic airway devices were used, i-gel size 1 and **l**aryngeal mask (standard **l**aryngeal mask, Solus, Intersurgical Ltd., Wokingham, Berkshire, UK) size 1. For neonatal simulation manikins, the Neonatal Resuscitation Model LM-089 (Koken Co., Ltd., Bunkyo-ku, Tokyo, Japan) and Laerdal Neonatal Intubation Trainer (Laerdal Medical., Stavanger, Norway) were used (Figure 1).

### 2.2. Study Procedures

First, the participants were provided with the study procedures and instructed on how to use the **l**aryngeal mask and the i-gel using printed instructions (Figure 1). Subsequently, a member of the study team, who was a qualified instructor in the neonatal cardiopulmonary resuscitation (NCPR) training course [16], demonstrated the insertion of each supraglottic airway device into both manikins. The water-soluble lubrication was used as instructed for each supraglottic airway device. All participants performed four insertion attempts using both types of supraglottic airway devices and manikins (Figure 1), the order of which was randomized using the RAND function of the spreadsheet software (Microsoft Excel 2013, Microsoft Corporation, Redmond, WA, USA). Two staff members observed each insertion attempt. The participants were instructed to declare the commencement of the insertion procedure, pick up the supraglottic airway device, and inform the staff when their attempts at insertion were complete. The time spent from the commencement of the procedure to its completion was recorded by a staff member of the study team. The study staff then connected the supraglottic airway device to a bag valve mask (Twenty-One Manual Resuscitator, Infants, GM Medical, Fukuoka, Japan) and manually ventilated it at a rate of approximately 60 times/min, according to the NCPR textbook [16]. When the bronchial pressure indicated on the built-in manometer showed elevations higher than 20 cmH_2_O more than six times in 10 ventilations, the attempt was considered successful. A video camera was used to record the procedure, along with the view of the manometer gauge, enabling video review in cases where the live assessment of the insertion time and success was ambiguous.

### 2.3. Backgrounds of Participants

We collected the background information of each participant, including specialty, years of healthcare service, previous experience using either type of neonatal supraglottic airway device, and past participation in an NCPR training course.

### 2.4. Data Analysis

All statistical analyses were performed using SPSS for Windows (ver. 29, IBM, Armonk, NY, USA). The primary outcome was the successful insertion of the supraglottic airway devices within an insertion time shorter than the 75th percentile of all successful attempts. The secondary outcomes were: (i) successful insertion regardless of the time spent, (ii) insertion time in all successful attempts, and (iii) successful insertion within an insertion time shorter than the 50th percentile of all successful attempts. A generalized estimating equation was used to investigate the dependence of outcomes on participants’ backgrounds by incorporating findings from repeated assessments of the same participants. Logistic regression and linear regression structures were used for the binary and continuous outcomes, respectively. A univariate analysis was performed to determine the crude effect of each independent variable. For the primary outcome, multivariate analysis was performed to clarify the dependence of successful insertion on the supraglottic airway device type, with adjustment for the specialty, career, and experience of the participants and the order of insertion. The threshold for significance was set at *p* < 0.05 and not corrected for multiple comparisons; however, *p*-values close to the threshold were interpreted cautiously. Values are shown as number (percentage), mean (standard deviation), median (quartile ranges), odds ratio (OR), and 95% confidence intervals unless otherwise noted.

## 3. Results

### 3.1. Study Population

Ninety-one nurses (*n* = 41), midwives (*n* = 15), physicians (*n* = 15), student nurses (*n* = 13), medical clerks (*n* = 6), and one hospital specialist participated in the study. Forty-two (46%) participants completed the NCPR training course. The number of years of healthcare service was 5 (2–11) (Table 1).

### 3.2. Total Successful Rate and Insertion Time

All participants successfully completed the study procedure, leaving data from 364 attempts available for analysis; no data were missing. Of the 364 insertion attempts made by the 91 participants, 200 (54.9%) were successful. The insertion time was 6.0 (10.7–16.4) s in the whole dataset and 6.5 (10.9–16.7) s when failed attempts were excluded.

### 3.3. Determinants of the Primary Outcome

In the univariate analysis, successful insertion < 16.4 s (75th percentile of all successful attempts) was negatively associated with the use of the Laerdal simulator (odds ratio [OR], 0.043; 05% 95% confidence interval [CI], 0.023–0.079; *p* < 0.001) and **l**aryngeal mask (OR, 0.619; 95% CI, 0.464–0.825; *p* = 0.001) (Table 2).

In the multivariate analysis adjusted for insertion order, occupation, and clinical experience, the outcome was negatively associated with the use of the Laerdal simulator (OR, 0.034; 05% 95% CI, 0.018–0.066; *p* < 0.001) and **l**aryngeal masks (OR, 0.474; 95% CI, 0.292–0.771; *p* = 0.003) (Table 3).

### 3.4. Determinants of Secondary Outcomes

When the time spent on insertion was ignored, successful insertion was negatively associated with the use of the Laerdal simulator (OR, 0.004; 95% CI, 0.002–0.011; *p* < 0.001) (Table 4).

When data from all successful attempts were assessed, the insertion time was associated with the duration of clinical experience (>11 years vs. 2–11 years; regression coefficient, 0.020; 95% CI, 0.001–0.602; *p* = 0.024) and the use of **l**aryngeal masks (regression coefficient, 151.091; 95% CI, 11.099–2056.846; *p* < 0.001) (Table 5).

Successful insertion within 10.9 s (50th percentile of all successful attempts) was inversely associated with the Laerdal simulator (OR, 0.099; 95% CI, 0.054–0.182; *p* < 0.001) and **l**aryngeal mask (OR, 0.196; 95% CI, 0.122–0.316; *p* < 0.001).

## 4. Discussion

Using two neonatal simulation manikins with different difficulties for supraglottic airway device insertion, we demonstrated that the i-gel was superior to the **l**aryngeal mask, with a higher success rate within a fixed period and shorter insertion time among all successful attempts. The success rate was not associated with healthcare specialty, years of healthcare service, previous experience using supraglottic airway devices, or participation in NCPR training. These data suggest the potential advantages of using the i-gel and may justify the relevance of conducting a randomized trial comparing **l**aryngeal masks and i-gels in neonatal resuscitation.

### 4.1. Comparison of Supraglottic Airway Devices in Airway Management

Since the first supraglottic airway device was proposed [6], several revised models have been created [7]. These newly developed devices are classified into those with air-inflatable cuffs and those with ergonomically preshaped cuffs. The i-gel is a relatively new and unique supraglottic airway device with a gel-filled cuff. Findings from studies comparing insertion times between these devices lack consistency. Lee et al. compared the **l**aryngeal mask (Laryngeal Mask Classic, Laryngeal Mask Company Ltd., Henley-on-Thames, UK) and i-gel in 99 children aged 1–108 months and demonstrated that the insertion time was shorter with the i-gel than with the **l**aryngeal mask [8]. Kayhan et al. reported similar findings in favor of i-gel versus **l**aryngeal masks (Laryngeal Mask ProSeal, LMA North America Inc., San Diego, CA, USA) in 50 infants weighing 2–5 kg [10]. However, a randomized equivalence trial in 170 children aged 3 months to 11 years showed no difference in the insertion time between the **l**aryngeal mask (Laryngeal Mask Supreme, LMA North America, San Diego, CA, USA) and the i-gel [9]. Another randomized study in 54 relatively smaller infants weighing <10 kg also noted no difference in the insertion time between the **l**aryngeal mask (classic laryngeal mask) and i-gel [11]. In contrast, a study of 60 randomized pediatric patients weighing between 10 and 25 kg found that the insertion time for the **l**aryngeal mask (Laryngeal Mask Supreme) was significantly shorter than that of the i-gel [12]. These studies investigated the performance of supraglottic airway devices using a wide age range of children during the induction of general anesthesia. However, few studies have focused on airway management during neonatal resuscitation. Using neonatal simulation manikins, our study demonstrated that using the i-gel was associated with more precise and faster airway management than the **l**aryngeal mask, regardless of the specialty and experience of the participating healthcare providers. These findings may justify the need for a randomized clinical trial to compare the performances of **l**aryngeal masks and i-gels in neonatal resuscitation.

### 4.2. Dependence of Successful Insertion on Providers’ Backgrounds

Thus far, few studies have investigated the dependence of successful supraglottic airway device insertion on the type of providers, presumably because most previous studies addressed supraglottic airway device-based airway management during general anesthesia, which is usually performed by trained anesthesiologists [8,9,10,11,12,13,14,15]. One study compared the success rates of airway management by nurse specialists, paramedics, and anesthesia residents using three different supraglottic airway devices in a neonatal simulation manikin (Ambu M-Mega Code Baby, Ambu, Hessen, Germany) [17]. This study found no significant differences in the success rates or airway management scores among the three health occupations. Our current study showed that the primary and secondary outcomes were not associated with specialty, years of healthcare service, previous experience using supraglottic airway devices, or participation in NCPR training, except that a longer duration of healthcare service showed a modest association with a shorter time to successful insertion. Our findings suggest that airway management using a **l**aryngeal mask and i-gel is minimally influenced by the provider’s background. The printed instructions and live demonstrations by the study team staff might have been sufficient to saturate the working image of the providers required to use these supraglottic airway devices. Indeed, in studies examining the efficacy of **l**aryngeal masks in neonatal resuscitation, only a simple additional module was used in basic and essential newborn resuscitation programs [4,18]. However, another explanation is possible. In this study, as well as with previous experience of using supraglottic airway devices and participation in the NCPR course, the order of attempts did not affect the success rate of the insertion. Furthermore, we observed a trend of decreasing success rate with an increasing number of attempts. Although the exact reason remains unclear, this may be explained by fatigue and loss of concentration resulting from multiple insertions. Airway management skills using supraglottic airway devices may be difficult to improve at a certain level. Training programs that provide extended skills for supraglottic airway device insertion are required.

### 4.3. Relevance of Evaluating Supraglottic Airway Devices in Simulation Manikins

This study was conducted in neonatal simulation manikins. Owing to this setting, participating providers were able to repeat insertion attempts several times, which helped highlight the dependence of successful insertion on the device type and provider’s backgrounds. To account for possible inter-neonate variations in the anatomical structure in the clinical setting, we used two different types of neonatal simulation manikins. While both manikins were designed to mimic term neonates weighing 2.5–3.5 kg and are widely used in Japanese NCPR training courses, the success rate of insertion remained at 9.9% for the Laerdal model and 72.0% for the Koken model. Several settings specific to the current study might have contributed to the relatively low success rate. First, since the Laerdal manikin is specifically designed for endotracheal intubation training, the tongue is relatively larger in size, leaving less space around the larynx compared to the Koken manikin (Figure 1). Thus, structural issues might have contributed to the lower insertion success rate, especially for the Laerdal manikin. Second, the participating providers were given only a brief instruction and demonstration before performing airway management, which might be improved with a number of appropriate practice sessions using the same devices. Third, although we used lubrication before insertion, its efficacy might differ between two SGA devices, and also between manikins and human neonates. Finally, we defined successful insertion as achieving the peak pressure of ≥20 cmH_2_O with a constant squeezing force of the bag provided by an experienced staff neonatologist. However, in clinical settings, the squeezing force can be adjusted in response to poor chest movement until sufficient ventilation is achieved. Therefore, the current definition of successful insertion might excessively be stringent, presumably identifying highly precise insertions of SGA devices with the minimum leakage of the air. Indeed, in a study using a neonatal manikin, failures in insertion attempt and airway management were defined as removal of the device from the mouth and a failure to achieve a successful insertion within the three attempts, respectively [19]. Another study using seven different SGA devices and a self-inflating bag in a neonatal manikin demonstrated that the mean peak inspiratory pressure of 20 cmH_2_O was achieved with only three devices [20]. As Micaglio et al. performed in their study of a neonatal manikin, different positive pressures may need to be applied in future studies to compare the level of oropharyngeal leak pressures between different SGA devices [21]. These fundamental differences between resuscitation simulating in future clinical studies are required to assess these findings in simulation manikins and human neonates need to be noted when interpreting our findings into clinical practice.

### 4.4. Limitaions

In addition to the significant difference between resuscitation manikins and human neonates, several other limitations of our study need to be considered. First, unlike most previous studies that targeted a relatively wide range of infants and children undergoing general anesthesia, our study focused on neonatal resuscitation. Subsequently, only size 1.0 **l**aryngeal mask and i-gel were used with neonatal resuscitation manikins, rendering comparisons with other studies difficult. Considering the increasing use of supraglottic airway devices in preterm infants, including those as young as 28 weeks gestational age and with birth weights as low as 810 g [22], further research using manikins designed to mimic preterm-born neonates is warranted. Second, this study recruited providers at a single center, resulting in relatively uniform backgrounds of the participating providers, especially in terms of their skills in neonatal resuscitation. Additionally, because of the limited study population, we were not able to fully assess the influence of providers’ backgrounds to the outcome.

## 5. Conclusions

Our study of 91 healthcare providers suggested that for the airway management of neonatal simulator manikins, using i-gel was associated with a higher success rate within a fixed time and shorter insertion time within successful attempts than that of the **l**aryngeal mask. Considering that the provider’s specialty, experience, and training had minimal impact on successful insertion, prospective clinical studies are warranted to investigate the potential advantage of using the i-gel for neonatal resuscitation by comparing laryngeal masks and i-gels.

## Figures and Tables

**Figure 1 children-11-00530-f001:**
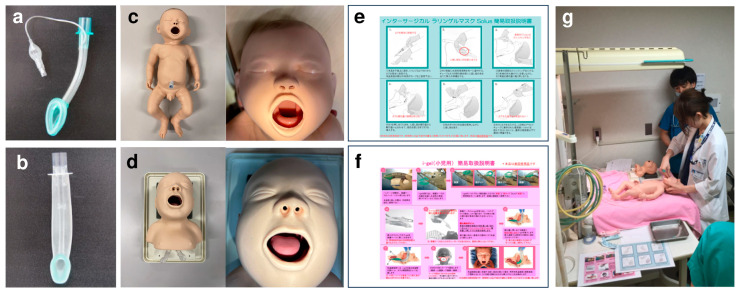
Supraglottic airway devices used and study procedure. Laryngeal mask (**a**), i-gel (**b**), Koken Neonatal Resuscitation Model LM-089 (**c**), Laerdal neonatal intubation trainer (**d**), one-page instructions for the laryngeal mask (**e**) and the i-gel (**f**) (original Japanese instructions see Appendix A), and actual study procedure using two types of manikins and two different supraglottic airway devices (**g**).

**Table 1 children-11-00530-t001:** Backgrounds of participating providers.

Specialty *	
Nurse	41 (45.1)
Midwife	15 (16.5)
Physician	15 (16.5)
Student nurse	13 (14.3)
Medical clerk	6 (6.6)
Hospital play specialist	1 (1.1)
Experience and training *	
Years of healthcare service	5 (2–11)
Participation to NCPR course	168 (46.2)
Experience of using supraglottic airway devices	
i-gel	20 (5.5)
Laryngeal mask	36 (9.9)
Any of above	40 (11.0)
Outcomes **	
Time spent in s	
All attempts	11 (6–16)
Successful insertion only	11 (1–82)
Successful insertion	
Any success	200 (54.9)
Success < 16.7 s	149 (40.9)
Success < 10.9 s	97 (26.6)
Success < 6.5 s	45 (12.4)

Values are from * 91 providers and ** 364 insertion attempts, shown as number (%) or median (quartile range). Abbreviation: NCPR, Neonatal Cardio-Pulmonary Resuscitation.

**Table 2 children-11-00530-t002:** Determinants of successful insertion within 16.7 s (univariable analysis).

	Success < 16.7 s		Odds Ratio			*p*
Variables	Yes (*n* = 149)	No (*n* = 215)	Mean	95% Confidence Interval		
				Lower	Upper	
Insertion order						
First	42 (46.2%)	49 (53.8%)	Reference			
Second	40 (44.0%)	51 (56.0%)	0.915	0.521	1.608	0.758
Third	36 (39.6%)	55 (60.4%)	0.764	0.426	1.369	0.365
Fourth	31 (34.1%)	60 (65.9%)	0.603	0.309	1.175	0.137
Specialty						
Physician	24 (40.0%)	36 (60.0%)	0.955	0.619	1.471	0.834
Other	125 (41.1%)	179 (58.9%)	Reference			
Years of healthcare service						
≤2	36 (39.1%)	56 (60.9%)	1.023	0.656	1.595	0.920
2–11	71 (38.6%)	113 (61.4%)	Reference			
>11	42 (47.7%)	46 (52.2%)	1.453	0.931	2.267	0.100
Participation to Neonatal Cardio-Pulmonary Resuscitation course						
Yes	70 (41.7%)	98 (58.3%)	1.058	0.724	1.546	0.771
No	79 (40.3%)	117 (60.0%)	Reference			
Experience of using i-gel						
Yes	8 (40.0%)	12 (60.0%)	0.96	0.588	1.566	0.870
No	141 (41.0%)	203 (59.0%)	Reference			
Experience of using laryngeal mask						
Yes	15 (41.7%)	21 (58.3%)	1.034	0.532	2.009	0.921
No	134 (40.9%)	194 (59.1%)	Reference			
Experience of using i-gel or laryngeal mask						
Yes	17 (42.5%)	23 (57.5%)	1.075	0.587	1.969	0.815
No	132 (40.7%)	192 (59.3%)	Reference			
Simulation manikin						
Laerdal	18 (9.9%)	164 (90.1%)	0.043	0.023	0.079	<0.001
Koken	131 (72.0%)	51 (28.0%)	Reference			
Type of supraglottic airway device						
Laryngeal mask	64 (35.2%)	118 (64.8%)	0.619	0.464	0.825	0.001
i-gel	85 (46.7%)	97 (53.3%)	Reference			

**Table 3 children-11-00530-t003:** Determinants of successful insertion within 16.7 s (multivariable analysis).

	Odds Ratio			*p*
Variables	Mean	95% Confidence Interval		
		Lower	Upper	
Insertion order				
First	Reference			
Second	0.625	0.287	1.359	0.236
Third	0.545	0.258	1.151	0.112
Fourth	0.434	0.192	0.982	0.045
Specialty				
Physician (vs. Othres)	0.843	0.24	2.963	0.790
Years of healthcare service				
≤2	1.065	0.458	2.475	0.884
2–11	Reference			
>11	2.003	0.908	4.418	0.085
Experience of using supraglottic airway device				
Yes (vs. No)	1.312	0.283	6.08	0.728
Simulation manikin				
Laerdal (vs. Koken)	0.034	0.018	0.066	<0.001
Type of supraglottic airway device				
Laryngeal mask (vs. i-gel)	0.474	0.292	0.771	0.003

**Table 4 children-11-00530-t004:** Determinants of successful insertion (univariable analysis).

			Odds Ratio			*p*
Variables	Yes (*n* = 200)	No (*n* = 164)	Mean	95% Confidence Interval		
				Lower	Upper	
Insertion order						
First	52 (57.1%)	39 (42.9%)	Reference			
Second	53 (58.2%)	38 (41.8%)	1.046	0.557	1.964	0.889
Third	50 (54.9%)	41 (45.0%)	0.915	0.493	1.697	0.777
Fourth	45 (49.5%)	46 (50.5%)	0.734	0.377	1.428	0.362
Specialty						
Physician	30 (50.0%)	30 (50.0%)	0.788	0.624	0.995	0.046
Other	170 (55.9%)	134 (44.1%)	Reference			
Years of healthcare service						
≤2	49 (53.3%)	43 (46.7%)	0.896	0.677	1.187	0.444
2–11	103 (56.0%)	81 (44.0%)	Reference			
>11	48 (54.5%)	40 (45.5%)	0.944	0.682	1.306	0.726
Participation to NCPR course						
Yes	92 (54.8%)	76 (45.2%)	0.986	0.77	1.264	0.913
No	108 (55.1%)	88 (44.9%)	Reference			
Experience of using i-gel						
Yes	10 (50.0%)	10 (50.0%)	0.811	0.711	0.924	0.002
No	190 (55.2%)	154 (44.8%)	Reference			
Experience of using laryngeal mask						
Yes	21 (58.3%)	15 (41.7%)	1.165	0.731	1.857	0.520
No	179 (54.6%)	149 (45.4%)	Reference			
Experience of using i-gel or laryngeal mask						
Yes	23 (57.5%)	17 (42.5%)	1.124	0.734	1.721	0.592
No	177 (54.6%)	147 (45.4%)	Reference			
Simulation manikin						
Laerdal	23 (12.6%)	159 (87.4%)	0.004	0.002	0.011	<0.001
Koken	177 (97.3%)	5 (2.7%)	Reference			
Type of supraglottic airway device						
Laryngeal mask	99 (54.4%)	83 (45.6%)	0.957	0.788	1.162	0.655
i-gel	101 (55.5%)	81 (44.5%)	Reference			

**Table 5 children-11-00530-t005:** Determinants of time spent for successful insertion (univariable analysis).

	Time (s)	Regression Coefficient			*p*
Variables	Mean (Standard Deviation)	Mean	95% Confidence Interval		
			Lower	Upper	
Insertion order					
First	12.4 (13.3)	Reference			
Second	14.8 (11.8)	2.448	−2.232	7.129	0.305
Third	14.2 (11.1)	1.821	−3.012	6.654	0.46
Fourth	13.3 (9.0)	0.897	−3.475	5.27	0.687
Specialty					
Physician	13.2 (13.8)	−0.645	−6.159	4.869	0.819
Other	13.8 (11.0)	Reference			
Years of healthcare service					
≤2	13.4 (13.6)	−1.64	−6.481	3.201	0.507
2–11	15.0 (11.5)	Reference			
>11	11.1 (8.3)	−3.918	−7.329	−0.508	0.024
Participation to Neonatal Cardio-Pulmonary Resuscitation course					
Yes	13.5 (10.6)	−0.461	−3.926	3.004	0.794
No	13.9 (12.1)	Reference			
Experience of using i-gel					
Yes	12.9 (3.9)	−0.855	−4.133	2.423	0.609
No	13.7 (11.7)	Reference			
Experience of using laryngeal mask					
Yes	13.0 (9.8)	−0.746	−5.236	3.745	0.745
No	13.8 (11.6)	Reference			
Experience of using i-gel or laryngeal mask					
Yes	13.0 (9.3)	−0.823	−5.005	3.359	0.700
No	13.8 (11.7)	Reference			
Simulation manikin					
Laerdal	11.0 (7.1)	−3.092	−6.584	0.400	0.083
Koken	14.1 (11.8)	Reference			
Type of supraglottic airway device					
Laryngeal mask	16.2 (8.5)	5.018	2.407	7.629	<0.001
i-gel	11.2 (13.3)	Reference			

## Data Availability

The data presented in this study are available on request from the corresponding author due to privacy restrictions and ethical reasons.

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
