# Peer review of "Dependence of Successful Airway Management in Neonatal Simulation Manikins on the Type of Supraglottic Airway Device and Providers’ Backgrounds"

_children, 2024, doi:10.3390/children11050530_

Round 1
Reviewer 1 Report
Comments and Suggestions for Authors
This is an excellent paper about an airway modality that has the potential to improve neonatal airway management particularly during resuscitation by less experienced providers. The newer SGA models such as the i-gel are also potentially an improvement over older models such as the LMA.
Introduction
Line 63 – This reference says “Where resources and training permit, we suggest that an SGA may be used in place of a face mask for newborn infants of ≥34 0/7 weeks’ gestation receiving intermittent PPV during resuscitation immediately after birth (weak recommendation, low-certainty evidence)”. I think the way it is written in this paper suggests that the reference suggests that SGA should be rather than may be used instead of the face mask interface.
It would be useful to discuss the size limitations of the curently available SGA devices and perhaps discuss the use in the literature of babies smaller than the recommended weight limits. It is also possible that smaller models for smaller infants may be developed in the future.
Methods
It would be helpful to talk about the neonatal resuscitation experience that the staff at this hospital are exposed to to provide a better idea of how much experience the participants have with this skill.
Was the verification procedure of placement of the SGA by participants followed by PPV given by a study team member tested prior to the study? Is it possible that there was successful placement but then the SGA was displaced during the procedure of attaching the device to the SGA and then providing the manual ventilations? This also may be less likely to be reflective of what happens in real situations in which the same team that is placing the SGA also attaches the ventilation device and provides the PPV.
It should be stated that the sample size was limited by the staff available at the single hospital if this was the case.
Is there a reason that the 2 different neonatal simulators were chosen? Was it anticipated that one may be more difficult than the other.
Was any lubrication used when placing the SGA in the models?
Discussion
It is mentioned but perhaps more specifics on the differences in the airways of the 2 neonatal simulators (size, shape, material, etc) could be provided in section 4.3.
The finding of difference in the models suggests that one or both may not be reflective of what would happen in human patients.
Conclusions
I do not think that the statement ending in “using i-gel for the resuscitation of newborns is recommended” (line 284-286) should be used as this study is limited by the fact that it was only performed on 2 neonatal simulators (with different results) and not done on human neonates. However, it does support the recommendation that similar clinical studies in human subjects are warranted.
Author Response
We appreciate the valuable suggestions provided by the reviewer from an expert point of view, which helped clarify and improve the quality of our manuscript. Here is a point-by-point response to your comments.
Comments by the reviewers and author rebuttals:
Reviewer 1
Reviewer comment:
This is an excellent paper about an airway modality that has the potential to improve neonatal airway management particularly during resuscitation by less experienced providers. The newer SGA models such as the i-gel are also potentially an improvement over older models such as the LMA.
- Author response to the reviewer:
We appreciate the valuable suggestions provided by the reviewer from an expert point of view, which helped clarify and improve the quality of our manuscript. Here is a point-by-point response to your comments.
Introduction
Reviewer comment:
Line 63 – This reference says “Where resources and training permit, we suggest that an SGA may be used in place of a face mask for newborn infants of ≥34 0/7 weeks’ gestation receiving intermittent PPV during resuscitation immediately after birth (weak recommendation, low-certainty evidence)”. I think the way it is written in this paper suggests that the reference suggests that SGA should be rather than may be used instead of the face mask interface.
- Author response to the reviewer:
We reflect that our original expression was an overstatement. We have revised to use more appropriate wording employed by the reviewer and in the guidelines.
(p.2, line 63-65).
Reviewer comment:
It would be useful to discuss the size limitations of the curently available SGA devices and perhaps discuss the use in the literature of babies smaller than the recommended weight limits. It is also possible that smaller models for smaller infants may be developed in the future.
- Author response to the reviewer:
We agree with your advice. We have added a discussion point regarding the minimum weight and gestational limitations in limitation section (p11, line 293-296).
Methods
Reviewer comment:
It would be helpful to talk about the neonatal resuscitation experience that the staff at this hospital are exposed to to provide a better idea of how much experience the participants have with this skill.
- Author response to the reviewer:
We agree with the suggestion to mention neonatal resuscitation experience. However, we did not obtain the exact number of neonatal resuscitation cases each participant had. Instead, we have added information on the size of our facility and details about the participants to the best of our knowledge. (p.3, line 97-102).
Reviewer comment:
Was the verification procedure of placement of the SGA by participants followed by PPV given by a study team member tested prior to the study? Is it possible that there was successful placement but then the SGA was displaced during the procedure of attaching the device to the SGA and then providing the manual ventilations? This also may be less likely to be reflective of what happens in real situations in which the same team that is placing the SGA also attaches the ventilation device and provides the PPV.
- Author response to the reviewer:
We fully understand the reviewer's concern. However, there were no issues observed when switching between the study member inserting the supraglottic airway device and another study member confirming the success of insertion before the study. Furthermore, the same method was applied to all participants, we believe bias in the results was minimal. Additionally, in clinical settings, it could be common for the intubator to hold the device while requesting another healthcare provider to operate the bag, requiring device stabilization with tape. From these points, we believe the reviewer's concern can be alleviated.
Reviewer comment: It should be stated that the sample size was limited by the staff available at the single hospital if this was the case.
- Author response to the reviewer:
We fully agree with the advice from the reviewer. This study recruited providers at a single center, resulting in relatively small, and uniform backgrounds of the participating providers, especially in terms of their skills in neonatal resuscitation. Additionally, because of the limited study population, we were not able to fully assess the influence of providers’ backgrounds to the outcome. This has been added in Limitation section (p.11, line 296-299).
Reviewer comment:
Is there a reason that the 2 different neonatal simulators were chosen? Was it anticipated that one may be more difficult than the other.
- Author response to the reviewer:
We chose these two simulation manikins with the aim of conducting research that reflects actual clinical scenarios by using multiple manikins with different characteristics mimicking human variation. This is addressed in Discussion section (p.10, lines 264-266).
Reviewer comment:
Was any lubrication used when placing the SGA in the models?
- Author response to the reviewer:
In our experiment, we used lubricant as instructed. We have added this to the text (p3, line 114-115).
Discussion
Reviewer comment: It is mentioned but perhaps more specifics on the differences in the airways of the 2 neonatal simulators (size, shape, material, etc) could be provided in section 4.3.
The finding of difference in the models suggests that one or both may not be reflective of what would happen in human patients.
- Author response to the reviewer:
We have added photographs of the actual manikins in Figure 1, but since the Laerdal manikin is specifically designed for endotracheal intubation training, the tongue is relatively larger in size, leaving less space around the larynx compared to the KOKEN manikin. We speculate that this contributed to the difference in success rates, though we could not anticipate this issue a priori. Prior to the study, multiple study members performed verification tests using the two types of manikins and two types of supraglottic airway devices, with no issues identified. This may be attributable to the fact that the study members involved in verification possessed adequate knowledge and experience in both positive pressure ventilation and insertion of supraglottic airway devices. We have addressed this point in the Discussion section 4.3. (p.11, line 272-276).
Conclusions
Reviewer comment:
I do not think that the statement ending in “using i-gel for the resuscitation of newborns is recommended” (line 284-286) should be used as this study is limited by the fact that it was only performed on 2 neonatal simulators (with different results) and not done on human neonates. However, it does support the recommendation that similar clinical studies in human subjects are warranted.
- Author response to the reviewer:
We totally agree with the reviewer's opinion. Therefore, we have revised to use more appropriate wording, as well as mentioning that conducting clinical research would be warranted. (p.11, line 304-307).
Reviewer 2 Report
Comments and Suggestions for Authors
In the manuscript “Dependence of Successful Airway Management in Neonatal Simulation Manikins on the Type of Supraglottic Airway Device and Providers’ Backgrounds” the authors offer a randomized trial comparing insertion success of neonatal simulators using two different types of supraglottic airway devices (LMA). The newer i-gel LMA was associated with a higher success rate of placement and shorter insertion time in the study.
I congratulate the authors for presenting a comparison of insertion success in neonatal simulators with a newer LMA modality. While the authors discuss many different limitations, I think the biggest is that this study was performed on simulators and therefore may not accurately reflect ease of placement. However, this study does provide baseline data that will be useful to construct a prospective randomized trial.
Specific Comments:
-Page 7, table 2: I find it interesting that incidence of a successful placement decreased as number of insertions increased. Do you have a proposed reason for this?
-Page 10, line 249: The overall success rate of placement in your study was around 50%, which is not a very high success rate for a rescue airway device. Potentially the reason you did not see a difference based on provider specialty or previous experience was either the simulator does not accurately reflect real-life placement of LMAs..
-Page 10, line 255: What is the difference anatomical airway standpoint between manikins? In intubation simulations, is there a difference between number of successful attempts?
-Page 11, line 267: Another limitation was education of devices and placement techniques. The participants only saw a device placement demonstrated before testing. Had they undergone practice attempts then been tested, there may have been different results.
-Page 11, line 284-286: I do not think that form this simulation study you can appropriately recommend i-gel for neonatal resuscitation. Especially, given the fact the successful placement only occurred 50% of the time. Further in-vivo studies would be needed for such a statement to be declared.
Author Response
We greatly appreciate the valuable suggestions provided by the reviewer from an expert perspective, which helped clarify and improve the quality of our manuscript. We completely agree that our study was performed in a simulation setting, not actual neonatal resuscitation, and we have critically revised this point at Limitation section and Conclusion section. Please find below a point-by-point response to your comments including this issue.
Comments by the reviewers and author rebuttals:
Reviewer 2
Reviewer comment:
In the manuscript “Dependence of Successful Airway Management in Neonatal Simulation Manikins on the Type of Supraglottic Airway Device and Providers’ Backgrounds” the authors offer a randomized trial comparing insertion success of neonatal simulators using two different types of supraglottic airway devices (LMA). The newer i-gel LMA was associated with a higher success rate of placement and shorter insertion time in the study.
I congratulate the authors for presenting a comparison of insertion success in neonatal simulators with a newer LMA modality. While the authors discuss many different limitations, I think the biggest is that this study was performed on simulators and therefore may not accurately reflect ease of placement. However, this study does provide baseline data that will be useful to construct a prospective randomized trial.
- Author response to the reviewer:
We greatly appreciate the valuable suggestions provided by the reviewer from an expert perspective, which helped clarify and improve the quality of our manuscript. We completely agree that our study was performed in a simulation setting, not actual neonatal resuscitation, and we have critically revised this point at Limitation section and Conclusion section. Please find below a point-by-point response to your comments including this issue.
Specific Comments:
Reviewer comment:
-Page 7, table 2: I find it interesting that incidence of a successful placement decreased as number of insertions increased. Do you have a proposed reason for this?
- Author response to the reviewer:
We also anticipated becoming proficient and improving insertion techniques through multiple insertions. However, as pointed out by the reviewers, although there was no significant difference, the success rate of insertion decreased. While we are unsure of the precise reason, this is presumed to be due to fatigue and loss of concentration resulting from multiple insertions. We have added this point in the Discussion section 4.2.(p.10, line 256-259).
Reviewer comment:
-Page 10, line 249: The overall success rate of placement in your study was around 50%, which is not a very high success rate for a rescue airway device. Potentially the reason you did not see a difference based on provider specialty or previous experience was either the simulator does not accurately reflect real-life placement of LMAs.
- Author response to the reviewer:
We have added the more information about our facility and participants (p3, line 97-102). We have also added the possibility of structural issues with the manikins in the Discussion section 4.3. (p.11, line 272-276).
Reviewer comment:
-Page 10, line 255: What is the difference anatomical airway standpoint between manikins? In intubation simulations, is there a difference between number of successful attempts?
- Author response to the reviewer:
We could not find any study comparing the successful tracheal intubation rates between the two manikins. We have added photographs of the actual manikins in Figure 1, but since the Laerdal manikin is specifically designed for endotracheal intubation training, the tongue is relatively larger in size, leaving less space around the larynx compared to the KOKEN manikin as shown in Figure 1. We speculate that this contributed to the difference in success rates. We have added this point to the Discussion section 4.3
(p.11, line 272-276).
Reviewer comment:
-Page 11, line 267: Another limitation was education of devices and placement techniques. The participants only saw a device placement demonstrated before testing. Had they undergone practice attempts then been tested, there may have been different results.
- Author response to the reviewer:
We agree with the reviewer's opinion. We have mentioned this in the Limitations section (p10, line 266-269).
Reviewer comment:
-Page 11, line 284-286: I do not think that form this simulation study you can appropriately recommend i-gel for neonatal resuscitation. Especially, given the fact the successful placement only occurred 50% of the time. Further in-vivo studies would be needed for such a statement to be declared.
- Author response to the reviewer:
We completely agree with the reviewer's opinion. Therefore, we have revised to use more appropriate wording, as well as mentioning that conducting clinical research would be warranted. (p.11, line 304-307). 
Reviewer 3 Report
Comments and Suggestions for Authors
Dear Author/s
It has been a pleasure to read your current paper.
The manuscript is very well structured and includes usefull information on airway management in neonates.
Your study is well designed and addresses useful topics in neonatal respiratory distress management. The only observation would be that, in the results part some detailes that results from SPSS analysis could be presented in a simplified manner (eg. Tables 2, 3, 4 and 5).
If accepted for publication, it may contribute to a better understanding of the benefits of the new proceure of i-gel. I also appreciated the extensive part regarding the participants, which is relevant for these type of intervention, and the details on the training for the new procedure. Your study demonstrates that i-gel procedure is a skill that is available for all levels of prior traingin of healthcare providers.
With all best wishes
Author Response
We appreciate the reviewer's valuable advice. We hope that our study results may help promote the conduct of clinical research in future.
Comments by the reviewers and author rebuttals:
Reviewer 3
Reviewer comment: It has been a pleasure to read your current paper.
The manuscript is very well structured and includes useful information on airway management in neonates.
Your study is well designed and addresses useful topics in neonatal respiratory distress management. The only observation would be that, in the results part some details that results from SPSS analysis could be presented in a simplified manner (eg. Tables 2, 3, 4 and 5).
- Author response to the reviewer:
As the reviewer pointed out, our Table indeed contains a substantial amount of information. However, we believe the presented information is essential to demonstrate the reliability of the data. After careful consideration, we have modified Table 3 by omitting some rows.
Reviewer comment:
If accepted for publication, it may contribute to a better understanding of the benefits of the new procedure of i-gel. I also appreciated the extensive part regarding the participants, which is relevant for this type of intervention, and the details on the training for the new procedure. Your study demonstrates that i-gel procedure is a skill that is available for all levels of prior training of healthcare providers.
With all best wishes
- Author response to the reviewer:
We appreciate the reviewer's valuable advice. Following the advice, we have provided information about the number of newborn infants admitted to our NICU annually and the number who received mechanical ventilation, as well as details about the recruited participants (p3, line 97-102). We hope that our study results may help promote the conduct of such clinical research in future.
Round 2
Reviewer 2 Report
Comments and Suggestions for Authors
I appreciate the author's responses, and I would accept the manuscript int he updated form.
Author Response
The authors thank the reviewer for evaluating our revised manuscript and recommending it for publication.
